

# Local investigation of the crystal electric field ground-state in CeCu(Sb,Bi)2 heavy fermions

**Davi Zau⋆, G. S. Freitas, P. G. Pagliuso and R. R. Urbano**

Gleb Wataghin Institute of Physics, University of Campinas - UNICAMP,
Campinas, São Paulo 13083-859, Brazil

⋆ davizau@ifi.unicamp.br

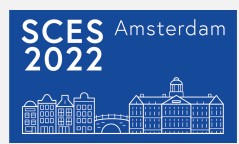

*International Conference on Strongly Correlated Electron Systems
(SCES 2022)
Amsterdam, 24-29 July 2022*

## Abstract

In this work, we performed systematic Nuclear Magnetic Resonance (NMR) and magnetic susceptibility experiments in $CeCuSb_2$ single-crystals. The main findings were compared to previous report for $CeCuBi_2$. [1] The NMR spectra and transferred hyperfine coupling for the $^{63}Cu$ nuclei were obtained aiming to observe their correlation with the crystal electric field (CEF) effects on the $Ce^{3+}$ ($J = 5/2$) multiplet. Besides, in an attempt to elucidate the magnetic structure through NMR measurements at different magnetic fields orientations, we observed a magnetic transition at $T \approx 8\,K$ higher than the Néel temperature $T_N$ measured by magnetic susceptibility indicating the development of short-range magnetic ordering above $T_N$. In addition, the wipe out of the main NMR resonance line and a persistent spin-echo signal throughout the whole frequency-swept range suggest the possibility of an incommensurate magnetic structure in $CeCuSb_2$. Furthermore, the small transferred hyperfine coupling constant found for $CeCuSb_2$ indicates a scenario with more localized $Ce^{3+}$ 4f electrons than for $CeMIn_5$ (M = Co,Rh,Ir) heavy fermions family. Additionally, subtle changes in the hybridization between the $^{63}Cu$ with the $4f^1$ $Ce^{3+}$ electrons in distinct magnetic field orientations allowed us to provide detailed information and map out the 4f CEF orbital ground-state of $CeCu(Sb,Bi)_2$ via NMR measurements.

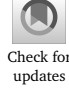
## 1 Introduction

In recent years, scientists put much effort into understanding unconventional superconductivity [2, 3] and several new families of complex superconductors were found [4, 5]. One particularly interesting and heavily studied class of these materials is the heavy fermions superconductors [6]. They had shown remarkable underlying physical properties, mainly due to the interplay between the Ruderman-Kittel-Kasuya-Yosida (RKKY) and Kondo interactions [7],

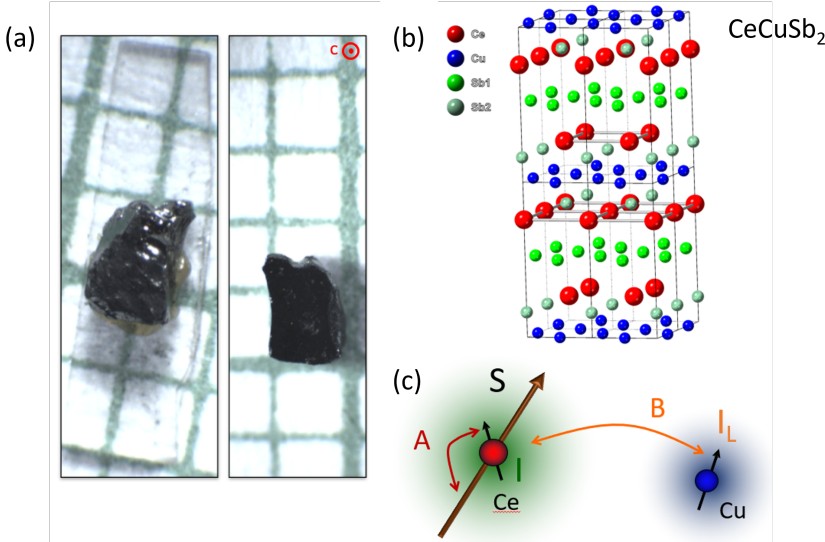

Figure 1: (a) CeCuSb$_2$ sample used in this study. (b) CeCuSb$_2$ structure. One can see that Cu and Sb are in different planes. (c) Transferred hyperfine coupling scheme. $A$ is the direct hyperfine coupling given by the interaction between the Ce$^{3+}$ 4f$^1$ electron spins ($S$) with the Ce nuclear spins (I). However, $I = 0$ for Ce nuclei. Therefore, $^{63}$Cu NMR is sensitive through the interaction of the Ce $S$ spins with the Cu nuclear spins $I_{Cu}$ through the transferred hyperfine coupling $B$.

which leads to a plethora of complex quantum condensed matter phenomena beyond superconductivity. Properties such as Non-Fermi liquid behavior, field-induced and quantum phase transitions, the coexistence of magnetism and superconductivity, and many others are not uncommon in such systems [8,9]. In this context, an interaction that showed to play an important role in the definition of the ground-state properties of these materials is the crystalline electric field (CEF) [10]. Recently, a study exhibited a direct relation between the magnetic and superconducting transition temperatures ($T_N$ and $T_c$, respectively) with the CEF effects for the Ce-115 family [11]. Moreover, a recent essay established a connection between the CEF and the transferred hyperfine coupling for the same compounds through nuclear magnetic resonance (NMR) [12].

Therefore, to observe whether the connection holds for other heavy fermion compounds such as CeCu(Bi,Sb)$_2$, we have performed systematic $^{63}$Cu NMR experiments ($I = \frac{3}{2}$, $\gamma_N = 11.285 \frac{MHz}{T}$) and magnetic susceptibility measurements in CeCuSb$_2$ single crystals. Through the Knight shift data, we directly extracted the transferred hyperfine coupling $B_{hf}$ via the Clogston-Jaccarino plot [13]. Moreover, within a mean-field framework, we obtained the Ce$^{3+}$ ($J = 5/2$) CEF parameters from magnetic susceptibility fittings [14]. Finally, comparing the results presented here with previous ones reported for CeCuBi$_2$ [1], we could evaluate the correlation between the CEF parameter and the hyperfine coupling in this family.

In addition, we have also attempted to elucidate the CeCuSb$_2$ magnetic structure anisotropy through NMR measurements, mainly in two field directions: perpendicular and parallel to the crystallographic $c$ axis. Employing careful analysis of the Knight shift in both directions, we obtained some hints regarding the orientation and the commensurability of the magnetic structure in the ordered state of CeCuSb$_2$.

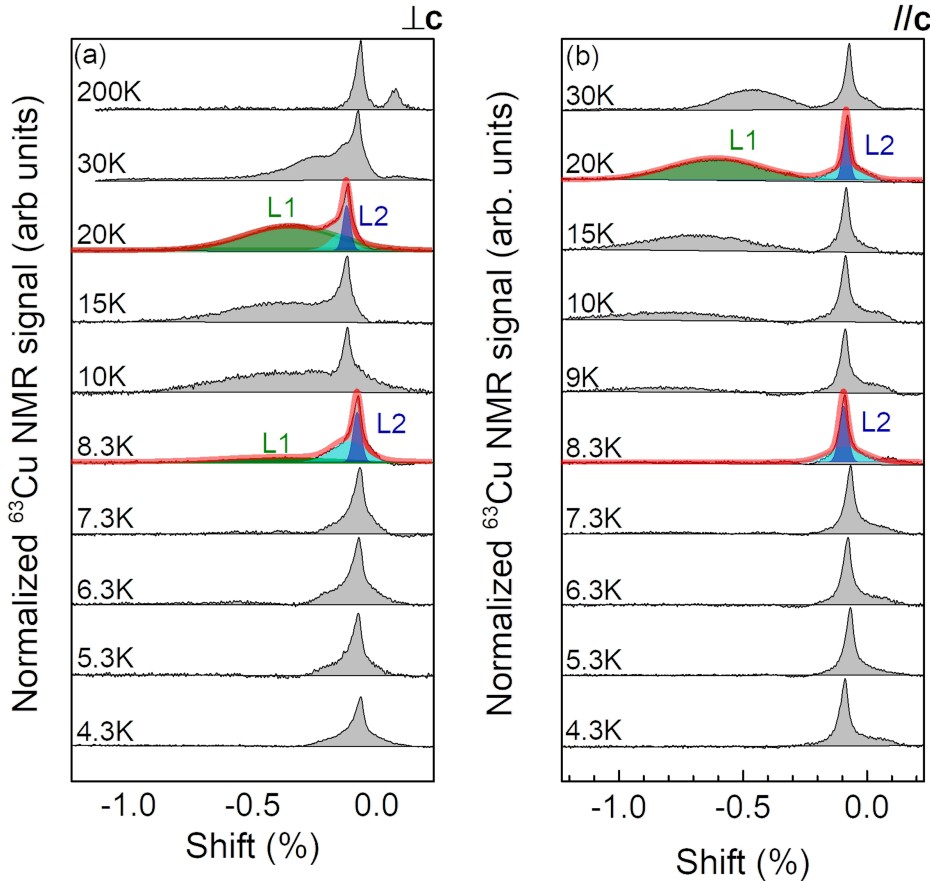

Figure 2: Normalized $^{63}$Cu NMR spectra in two distinct magnetic field orientations: (a) $H \perp c$ and (b) $H \parallel c$ at different temperatures. One can see mainly two distinct resonance lines namely L1 and L2, indexed as shown. L1 is strongly shifted, much broader, and disappears below $T \approx 8.3\,K$. L2 is slightly shifted, thinner, and seems to be less affected by temperature variation.

## 2   Methods

CeCuSb$_2$ single crystals (Fig 1.a) were obtained via the Sb self-flux method and the crystallographic structure was verified and reported in [15]. We carried out the magnetic susceptibility measurements using a commercial Superconductor Quantum Interferometer Device (SQUID) at $7\,T$ along the c-axis and in the ab-plane.

We have performed the NMR measurements using a high homogeneity superconducting magnet with a variable $12.1\,T$ field in a Helium-4 cryostat. A radio-frequency coil was manufactured with silver wire and set to be swept within the resonance frequency between $70\,MHz > \nu > 80\,MHz$. The frequency-swept $^{63}$Cu NMR spectra were obtained by step-wise summing the Fast Fourier Transform of the spin-echo signal.

## 3   Results and Discussion

The NMR spectra as a function of temperature in both field orientations (perpendicular and parallel to the crystallographic $c$ axis) are shown in Figure 2 where one can observe some striking features: There are mainly 2 resonance lines indexed in Figure 2. The broader and strongly

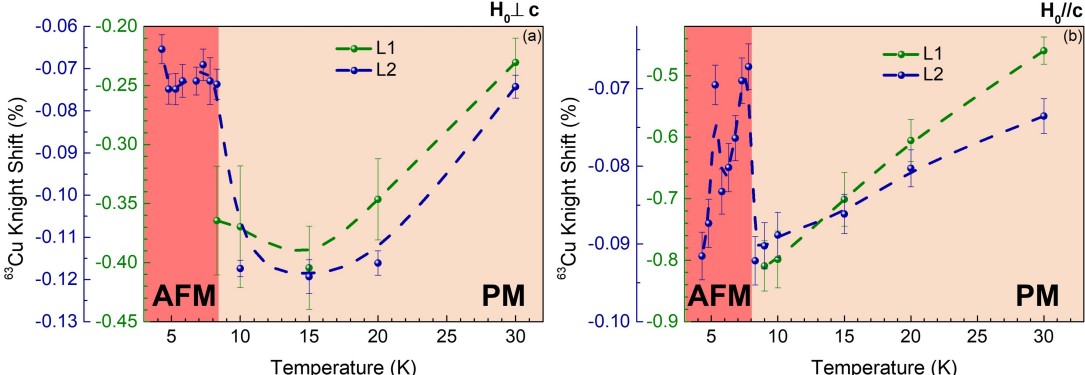

Figure 3: $^{63}$Cu NMR Knight Shift data for the spectra of Figure 2. One can see a clear change in the Knight Shift behavior at $T \leq 8.3$K for both orientations, indicating the onset of the antiferromagnetic transition.

shifted line, L1, disappears at low temperatures where the onset of the magnetic order kicks in. This leads us to assign L1 as the main resonance line in CeCuSb$_2$. The other resonance, L2, is narrower and slightly shifted and was attributed to $^{63}$Cu sites near intermetallic vacancies present in the crystal. In order to get the best overall NMR spectral fitting we have also considered another line, L2$^*$, in the same inhomogeneous environment as those $^{63}$Cu of L2 near the vacancies. However, since L2 and L2$^*$ behave virtually on the same way, we omitted the results for L2$^*$, for simplicity.

We have extracted the Knight shift shown in Figure 3 through best Gaussian fits and subsequent simulation combining all resonance signals. It is clear that all resonances probe the onset of an antiferromagnetic transition near $T \cong 8$ K. This is unexpected since the magnetic susceptibility measurements showed a $T_N \cong 5.8$ K indicating that some short-range magnetic order settles down before the long-range antiferromagnetic ordering [15]. Furthermore, although unclear in Fig.2, we observed a persistent spin-echo signal in the whole measured frequency range below 8 K, which suggests a possible incommensurate magnetic structure for CeCuSb$_2$. Additionally, as mentioned above, the main $^{63}$Cu NMR resonance signal disappears below the transition temperature, avoiding us from determining the magnetic structure of our sample. This complex magnetic structure could also explain the suppression of $T_N$ when compared to CeCuBi$_2$ [1], since it could lead to a more unstable magnetic ordering only able to settle down at lower temperatures. This corroborates with the rising magnetic frustration illustrated by the increase of the magnetic frustration parameter $\left( \frac{|\theta_{CW}|}{T_N} \right)$ previously reported in Ref [15].

We also measured the magnetic susceptibility at the same magnetic field used in the NMR measurements in order to obtain the hyperfine coupling constant. This can be accomplished through the Clogston-Jaccarino plot of the Knight shift as a function of the susceptibility shown in Figure 4. Thus, from the slope of the data, one may directly extract the hyperfine coupling constant $B_{hf}$ for both field orientations in CeCuSb$_2$.

In Table 1, we present the hyperfine coupling constants for each resonance observed in CeCuSb$_2$ and the CEF parameter $\alpha$ from the literature [1,15], which characterizes the degree of mixing between the $J_z$ manifolds and is directly related to the spatial anisotropy of the 4f CEF orbital as detailed in Ref [12]. In other words, $\alpha$ is the spin $J = 5/2$ contribution to the ground state wavefunction of the 4f$^1$ Ce$^{3+}$ electrons as illustrated in Fig. 5. The results for $B_{hf}$ of CeCuBi$_2$ from [1] are also presented in Table 1.

The CEF ground-state scheme for CeCuBi$_2$ and CeCuSb$_2$ is shown in Figure 5. Previous results [12] suggest that the transition metal hybridization with Ce correlates well with the 4f CEF orbital shape and that a ground state wave function with larger $\pm |5/2\rangle$ than the $\pm |3/2\rangle$

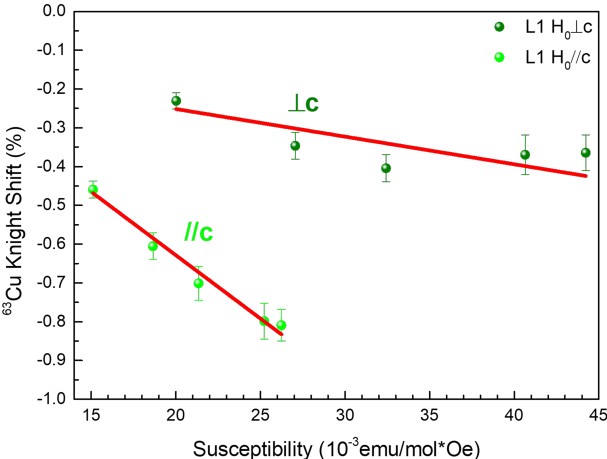

Figure 4: Clogston-Jaccarino plot for the $^{63}$Cu NMR data with the magnetic field $H_0 = 6.85\,T$ applied perpendicular and parallel to the crystallographic c-axis. The data were fitted (linear regression) with the equation $K = C\chi_{DC}^{mag} + K_0$, where $C = B_{hf}/(N_A\mu_B)$ with $B_{hf}$ as the transferred hyperfine coupling, $N_A$ as the Avogadro's number and $\mu_B$ as the Bohr magneton, $K_0$ is the temperature independent contribution to the Knight shift. The results we obtained for $C$ were $0,33(2)(molOe)/emu$ and $0,07(8)(molOe)/emu$ for the field parallel and perpendicular to the c axis respectively. The anisotropy is clearly shown by the distinct slopes.

contribution indicates a higher hybridization in the Cerium plane. However, although we see a drastic change in the CEF parameters for CeCuSb$_2$, there is no significant change in the $^{63}$Cu hybridization when the magnetic field is applied on the $a-b$ plane. Besides, if one compares the hyperfine coupling values with those obtained for the Ce-115 compounds [12], it is easy to realize a rather reduced energy scale for the Ce-112 compounds further corroborating with a lower hybridization between $^{63}$Cu and the Ce$^{3+}$ 4f$^1$ electrons in this latter case.

Nonetheless, a clear increase of the hyperfine coupling is noticed for the main resonance line L1 when comparing the data for distinct magnetic field orientations. We thus claim that this hyperfine coupling enhancement might be related to a change in the magnetic moment orientation. This is supported by the shift in the easy axis from parallel to perpendicular to the crystallographic $c$ axis observed by magnetic susceptibility measurements in Ce-112 [15].

Table 1: Transferred hyperfine coupling constants and $|\pm5/2\rangle$ spin ground state contribution ($\alpha$) as illustrated in Fig 5. Here, $B^{\parallel}$ ($B^{\perp}$) stands for the measurements done with the external field $H_0 \parallel c$ ($H_0 \perp c$). One can notice a drastic change in the 4f CEF parameter $\alpha$ not probed by the hyperfine coupling for $H_0 \perp c$. The missing value for CeCuBi$_2$ is due to the metamagnetic transition near 6 T in that magnetic field orientation.

|  | L1 | L2 | CeCuBi$_2$ |
|---|---|---|---|
| $\alpha$ | 0.43 | 0.43 | 0.98 |
| $B^{\parallel}(kOe/\mu_B)$ | 2.1(2) | 0.08(1) | - |
| $B^{\perp}(kOe/\mu_B)$ | 0.6(3) | 0.02(1) | 0.7(1) |

(a) $\Gamma_6 = |\pm 1/2\rangle$

$\Gamma_7^1 = 0.2|\pm 5/2\rangle + 0.98|\mp 3/2\rangle$

$J_T = \dfrac{5}{2}$

$\Gamma_7^2 = 0.98|\pm 5/2\rangle - 0.2|\mp 3/2\rangle$

(b) $\Gamma_7^2 = 0.89|\pm 5/2\rangle - 0.43|\mp 3/2\rangle$

$\Gamma_6 = |\pm 1/2\rangle$

$J_T = \dfrac{5}{2}$

$\Gamma_7^1 = 0.43|\pm 5/2\rangle + 0.89|\mp 3/2\rangle$

CeCuBi$_2$

CeCuSb$_2$

Figure 5: Scheme of the CEF ground state for the $4f^1$ Ce$^{3+}$ electrons for (a) CeCuBi$_2$ and (b) CeCuSb$_2$ with their respective orbital representations. Here, $\alpha$ is defined as the $|5/2\rangle$ contribution to the ground-state, where $\alpha = 0.98$ and $0.43$ for the CeCuBi$_2$ and CeCuSb$_2$, respectively. One can see that the ground-state orbital is more planar in the Bi-based compound, in contrast with that for the Sb-based one, suggesting a plausible enhancement of the transferred hyperfine coupling for the latter case, since the Cu nuclei are not in the same crystallographic plane as Ce.

Therefore, this demonstrates that NMR is sensitive to such a change, and allows us to define the configuration of the 4f CEF ground-state orbital in the structure, although a complete set of magnetic field orientations data is required to confirm this claim for the CeCuBi$_2$ sample. Measuring the magnetic moment orientation (magnetic structure) of these compounds and correlating it with the 4f CEF ground-state orbital would also bring new insight to this scenario.

## 4 Conclusion

In conclusion, we were able to probe the 4f$^1$ CEF ground-state orbital for this sample through $^{63}$Cu NMR investigations. Also, our study pointed out the possibility of an incommensurate magnetic structure for CeCuSb$_2$ mainly due to the persistent spin-echo signal observed below $T_N$ in the whole frequency swept range. The low values of hyperfine coupling for both CeCuSb$_2$ and CeCuBi$_2$ samples indicate a quite localized scenario for the 4f$^1$ Ce electrons if compared with the Ce-115 family. Moreover, we observed a drastic change in the hyperfine coupling constant as a function of magnetic field orientation, which might be related to the actual spacial orientation of the Ce$^{3+}$ ($J = 5/2$) CEF ground-state in the crystal structure. Therefore, we conclude that NMR is a suitable technique ideal to map the CEF ground-state orbital distribution in heavy fermion materials.

## Acknowledgements

This work was performed under the auspices of the Coordenação de Aperfeiçoamento de Pessoal de Nível Superior - Brasil (CAPES) - Grant # 88887.352079/2019-00, by the Conselho Nacional de Pesquisa e Desenvolvimento Tecnológico (CNPQ) - (Grants 304496/2017-0, 309483/2018-2, 314587/2021-7, 161490/2021-2) and by the Fundação de Amparo a Pesquisa do Estado de São Paulo (FAPESP) (Grants number: JP 2012/05903-6, 2016/14436-3, T 2017/10581-1, 2018/11364-7, 2019/26247-9)

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
