# Peer review of "Local investigation of the crystal electric field ground-state in CeCu(Sb,Bi)2 heavy fermions."

_SciPost Physics Proceedings, doi:SciPost Phys. Proc. 11, 017 (2023)_

## Round 1 · Referee Report · Anonymous (Referee 1) · 2022-11-29

Report

In the manuscript Zau et al. present NMR measurement to determine the hyperfine coupling constant in CeCuSb2. Its small value points to a localized nature of the f-electrons. The data are discussed in terms of the CEF ground state.

The paper reports new results and advances our knowledge of CeCuSb2 and related compounds. With suitable modifications it can be accepted for SciPost. Below I list some concerns, as well as suggestions to improve the paper.

  1. In Fig. 2 the main results of the paper are presented: NME spectra. Several (broad) peak are observed, but only L1 and L2 get labels. Not everything is clear: L1 for field perpendicular to c is broad and seems to consist of 3 subpeaks. (e.g. at 20 K). Please explain. How is its value determined? What are the peaks with a positive Knight shift. Explain this to the non-expert reader.

  2. The Clogston-Jaccarino (not Jacarinno) plot is fine, from this the hyperfine coupling constant is determined. But I find the ensuing discussion about Table 1 and Fig. 5 unclear. Please be more precise.

  3. I also do not get the importance of Fig.6. The left 2 orbital states with alpha = 0.43 and 0.47 are almost identical. Then for alpha = 0.47 the most right diagram has a tilted ground state orbital. Why?

  4. The authors should have their manuscript checked for English grammar, and correct typo’s. For instance in the abstract singlecrystals, eletric field, incomensurate. In Methods spectrum were obtained should read spectra were obtained. On the bottom of page 3: as Figure 2 should read in Figure 2. On page 4, probes should read probe, unexpecting should read unexpected. Etc.

  5. Figure 7 can be deleted.

  • validity: -
  • significance: -
  • originality: -
  • clarity: -
  • formatting: -
  • grammar: -

Author:  Davi Zau  on 2023-04-04  [id 3547]

(in reply to Report 1 on 2022-11-29)

We thank the referee for the valuable report. We revised the entire data set and prepared a summary of the changes with a proper explanation for each question raised in the attached pdf file.

Attachment:

Rebbutal_Scipost_April04_2023.pdf

---

## Round 2 · Author Response

Dear Editor,
Hereby we resubmit the manuscript Unveiling the 4f electrons hybridization in the CeCuSb2 heavy fermion, by Davi Zau, G.S. Freitas, P.G. Pagliuso, and R. R. Urbano for your reconsideration. We are grateful to the Referee for his thorough revision and appreciation of the great effort to get and analyze such experimental data as well as for his positive evaluation of our manuscript. In this revised version we rephrased some parts of the discussion and corrected the grammar in order to comply with all necessary modifications. We strongly believe that the manuscript in its revised version is both technically accurate and of sufficient significance to merit publication. Sincerely yours, Davi Zau on behalf of all co-authors.

---

## Round 2 · List of Changes

We reanalyzed all the data to eliminate any FFTsum artifacts. In fact, the graphs are now in a version much clearer to the reader. However, the overall results did not change.

We rephrased some of the discussion principally the discussion about the relation between the hyperfine coupling and the ground state CEF parameters.

We changed the title to best suit the conclusions to: Local investigation of the crystal electric field ground-state in CeCu(Sb,Bi)2 heavy fermions.

We corrected the grammar.

---

## Editorial Decision

published